# Self-reported COVID-19 severity among persons with tuberculosis infection in western Kenya, 2021

Hellen C. Barsosio[1,2], Brian Tangara[1], Joshua Ongalo[1], Morine Achieng[1],
Tegwen Marlais [3], Kimberly D. McCarthy[4], Kephas Otieno[1], Miriam Wanjiku[1],
Julian Matthewman[3], David Allen[3], Luke Hannan[2], Anand Date[5], Maia Lesosky[2],
Simon Kariuki[1], Aaron M. Samuels[2,4], Chris Drakeley[3], Feiko O. ter Kuile[1,2],
Taraz Samandari[4]*

**1** Center for Global Health Research, Kenya Medical Research Institute, Kisumu, Kenya, **2** Department of Clinical Sciences, Liverpool School of Tropical Medicine, Liverpool, United Kingdom, **3** Department of Infection Biology, Faculty of Infectious and Tropical Diseases, London School of Hygiene & Tropical Medicine, London, United Kingdom, **4** Division of Global HIV & TB (DGHT), Global Health Center (GHC), U.S. Centers for Disease Control & Prevention (CDC), Kisumu, Kenya, **5** DGHT, GHC, U.S. Centers for Disease Control & Prevention (CDC), Atlanta, Georgia, United States of America

* t_samandari@yahoo.com

## Abstract

Whilst a quarter of the world's population is estimated to be infected with *Mycobacterium tuberculosis*, it is unknown whether TB infection (TBI) increases the risk of severe COVID-19, which is relevant in TB-endemic settings, especially where HIV co-infection is also common. A convenience cohort of symptomatic and asymptomatic COVID-19 patients aged 8–80 years in western Kenya was followed daily for 14 days to assess disease progression using the validated inFLUenza-Patient-Reported-Outcome Plus signs and symptom tool. Nasal swabbing for SARS-CoV-2 was conducted to confirm the virus using polymerase chain reaction. QuantiFERON-TB Gold Plus was used to diagnose TBI. HIV status was based on self-reports. Between January 3, 2021, and January 20, 2022, 373 out of 387 participants had conclusive QuantiFERON results. At baseline, 5.9% (22/373) had self-reported severe COVID-19, 33.2% (124/373) had TBI, and 11.1% (38/341) reported being HIV-infected. Median follow-up of the cohort was 105 days (range 0–368). Self-reported severe COVID-19 was experienced by 10 of 124 (8.1%) participants compared with 12 of 249 (4.8%) without TBI (odds ratio [OR] 1.73, 95% CI 0.73-4.12, p = 0.21). HIV was not associated with self-reported severe COVID-19 (OR 3.13, 0.96-8.77, p = 0.039, adjusted OR 2.77, 95%CI 0.84-7.93, p = 0.070), but age ≥ 50 years was associated with self-reported severe COVID-19 (OR 3.73, 1.47-9.07, p = 0.004, adjusted OR 2.91, 95%CI 1.02-7.69, p = 0.035). One participant died of COVID-19 three days after diagnosis, and another participant developed active TB 128 days after his COVID-19 diagnosis and was successfully treated. Both were QuantiFERON positive.

**Data availability statement:** Data from this analysis is available as an Excel dataset: Barsosio, Hellen (2025), COVID-19 severity among persons with TB infection in Kenya, https://doi.org/10.57978/b8a3-ck29.

**Funding:** This study was supported through two grants from the Bill and Melinda Gates Foundation, INV-017985 (FtK) and INV-019400 (CD). This study received in-kind support from the President's Emergency Plan for AIDS Relief (PEPFAR) through U.S. Centers for Disease Control and Prevention (CDC). The funders had no role in study design, data collection and analysis, decision to publish, or preparation of the manuscript.

**Competing interests:** The authors have declared that no competing interests exist.

Self-reported severe COVID-19 was associated with older age and not TBI. Our finding that increased age was associated with self-reported severe COVID-19 is consistent with findings in multiple settings around the world.

## Introduction

Prior to the emergence of the COVID-19 pandemic, tuberculosis (TB) was the world's leading infectious disease killer [1]. As the spread of severe acute respiratory syndrome coronavirus 2 (SARS-CoV-2) rapidly swept the globe, TB public health practitioners, clinicians and researchers had concerns about the interactions between two potentially deadly respiratory pandemics with overlapping symptoms. Moreover, as a quarter of the world's population is estimated to be infected with the TB bacterium, the scope of the challenge remains of great interest [2].

Firstly, health systems disruptions precipitated by the COVID-19 pandemic reduced access to TB diagnostic services and disrupted programs for the provision of anti-TB treatment [3]. Secondly, higher hazards of COVID-19 deaths have been observed in patients with active TB disease as well as among those with a history of previous tuberculosis as demonstrated in the Western Cape province of South Africa, Pakistan and the Philippines [4], confirming what was previously observed in case series and reviews [5,6]. Thirdly, it has been speculated that immunomodulatory treatments for COVID-19 – particularly after the widespread use of corticosteroids – may increase the risk of individuals with TB infection (TBI) progressing to active TB disease [7], though evidence for the latter has been inconclusive [8–10].

A fourth concern is the interaction between COVID-19 and TBI. Some TB researchers have argued that persons with TBI may develop active TB after severe COVID-19 [11,12]. while others have suggested that in countries where TBI is common, severe COVID-19 is less likely [13].

Kenya is a high TB burden country [3]. In its 2020 Annual Report, the Ministry of Health estimated 140,000 TB cases with a 6.3% death rate [14]. In the two decades after 2000, Kenya had made steady progress in TB treatment completion and saw a decline in TB incidence, but 2020 and 2021 saw reversals in these gains as did much of the rest of the world [3]. The first case of COVID-19 was reported in Kenya on March 13, 2020. The disease rapidly spread nationwide in successive waves, particularly after the introductions of the alpha and delta variants of concern (VOC) between March and September of 2021 [15].

The present study explored whether SARS-CoV-2-infected individuals with TBI had a higher risk of self-reported severe COVID-19 in a TB-endemic setting. Uniquely, our study was conducted prospectively after the diagnosis of SARS-CoV-2 infection in a region of Africa with a high prevalence of HIV.

## Methods

### Study population and enrolment criteria

Participants were enrolled between January 3, 2021, and January 20, 2022, into the parent MALCOV study in western Kenya. MALCOV was an observational cohort

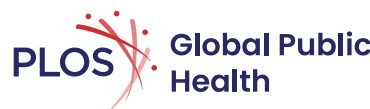

study conducted in six sites – five in Kenya and one in Burkina Faso – with high levels of malaria transmission with the chief objective of determining the effect of malaria and malaria treatment on SARS-CoV-2 viral clearance and COVID-19 disease progression. The TBI sub-study was exclusively conducted in western Kenya. During this time, the alpha and beta VOC were initially prevalent, but by the end of the second quarter of 2021, the delta VOC was predominant in western Kenya – peaking in June 2021 – followed by the omicron VOC in December of the same year. Participants were recruited into this convenience cohort from rural and urban areas of Kisumu (74%) and Siaya (26%) counties, respectively. From an unrelated concurrent study in neighboring rural Asembo area, population-based surveillance showed SARS-CoV-2 seroprevalence increased from 25% in March 2021 to 50% by July 2021 and 81% in July 2022 (Munywoki personal communication). A survey conducted in the Manyatta informal settlement of Kisumu City in July 2022 showed 87.5% seroprevalence among unvaccinated individuals (ibid.).

Enrolment criteria were age over 6 months, a positive test result for SARS-CoV-2 with either a polymerase chain reaction (PCR) or a rapid antigen detection test (RDT) plus being either (1) a patient attending COVID-19 test centers or clinics testing for COVID-19, or (2) admitted to the COVID-19 isolation or quarantine facilities, or (3) a contact of a confirmed or suspected case as part of contact tracing, or (4) among those targeted for testing due to the high-risk nature of their work (i.e., frontline healthcare workers). Study recruiters worked with local government focal persons in charge of community health workers to recruit potential participants at these facilities. Exclusion criteria included inability or unwillingness to give consent or anticipated by the study recruiter to be unable to adhere to the follow-up schedule.

After SARS-CoV-2 diagnosis, participants were assessed daily for fourteen days for influenza-like symptoms using the inFLUenza Patient-Reported Outcome Plus score (FLUPRO+), which uses a five-point scale. The FLUPRO and FLU-PRO+ tools have been demonstrated in several studies – that have included both ambulatory and hospitalized patients in multiple countries – to be reliable, valid, internally consistent and reproducible for persons ill with SARS-CoV-2, respiratory syncytial virus and influenza-like illness including in children [16–18]. The youngest participant in our cohort was 8 years old. The self-reported severity of COVID-19 illness was self-reported based on the FLUPRO+ reports. If the participant felt that they had severe symptoms at any time within the initial 14 days of daily recordkeeping after diagnosis, the participant was accounted as having experienced "severe COVID-19." Additional follow-up occurred for up to 12 months. Study recruiters maintained contact with participants initially through daily phone calls in the first two weeks, then by using text messaging in order to remind them of scheduled visits at study clinics. Appointment cards were provided to participants to show them their upcoming visits, and a participant diary was provided for the participant to record his/her symptoms.

Sociodemographic status and known chronic diseases, including HIV status, were provided by self-report. Medications received by participants were also recorded.

## Laboratory assays

Nasal swabs for SARS-CoV-2 were processed by extracting RNA using MagMaxTm Viral/Pathogen nucleic Acid Isolation Kit (ThermoFisher Scientific Inc, US) in a KingFisher Flex system (ThermoFisher Scientific Inc, US). Then, real-time polymerase chain reaction (RT-PCR) was done using the Novel Coronavirus (2019-nCoV) nucleic acid diagnostic kit (Sansure Biotech Inc, Changsha, People's Republic of China) in a 7500 Fast RT-PCR system (ThermoFisher Scientific Inc, US). The rapid antigen detection kit assay, NowCheck COVID-19 Ag-RDT (Bionote, Hwaseong-si, Korea), was also used as described in Onsongo et al [19].

A sample of blood from a finger prick was placed on a malaria rapid diagnostic test (First Response Malaria pLDH/HRP-2 Ag combo, Premier Medical Corporation Ltd., Gujarat, India, Catalogue No: I16FRC25) and a blood smear was prepared for malaria microscopy and read by two microscopists. A third microscopist resolved discrepant results [20]. A positive test was based on either a rapid diagnostic test or microscopy results.

Five milliliters of whole blood were collected in lithium heparin tubes, transported at room temperature and subsequently placed in QuantiFERON-TB Gold Plus (QFT, Qiagen, Germantown, MD) tubes and processed in the Kenya



Medical Research Institute's Centre for Global Health Research (KEMRI/CGHR), Tuberculosis Research Laboratory per manufacturer's guidelines. All QFT-positive participants were referred to a government facility for GeneXpert testing to rule out active TB disease.

### Statistical analyses

The chi-square test was used to compare proportions using Epi Info 7.2.4.0 (CDC, Atlanta GA). P-values <0.05 were considered statistically significant. Odds ratios were calculated, and multivariable logistic regression analyses of several models were conducted using R version 4.0.3 (R Foundation for Statistical Computing, Vienna, Austria). Variables showing an association with self-reported severe COVID-19 at a p-value below 0.05 in univariate analysis were included in the final multivariable model. Additional models were explored by including variables from univariate analysis with a cutoffs of p<0.1, p<0.2 and p<0.3.

### Ethics statement

The protocol was approved by KEMRI's Scientific and Ethical Review Unit and the Research Ethics Committee of the Liverpool School of Tropical Medicine. The U.S. Centers for Disease Control and Prevention's institutional review board relied on KEMRI. Participants of all ages were eligible. Those 18 years and over provided written informed consent; parents or guardians provided consent for children, and children between the ages of 13–17 provided assent. The parent MALCOV study was registered on clinicaltrials.gov (NCT04695197).

### Results

We screened 3,547 persons, among whom 628 were SARS-CoV-2 infected and met enrollment criteria (Fig 1). Reasons for SARS-CoV-2 testing were: COVID-19 suspects (45.0%) who were tested at clinics, contacts of known cases (15.5%), asymptomatic screening at local clinics (7.2%) and a variety of other reasons (32.0%) such as having a high-risk occupation.

A QFT test was conducted on blood samples from 387 (61.6%) SARS-CoV-2 positive participants; 241 (38.3%) did not receive a QFT test due either to the lack of QFT kits at the time of their enrolment or because they were enrolled at locations too distant to begin QFT tube 37°C incubation within the manufacturer's maximum of 16 hours from sample collection. As Kenya's national vaccination campaign had only recently begun, only five study participants were known to have received a COVID-19 vaccine prior to enrolment which was also at the time of SARS-CoV-2 diagnosis.

Because they had indeterminate QFT results, 14 of 387 (3.6%) participants were excluded from the main analysis; the proportion of individuals reporting severe COVID-19 symptoms was significantly higher among those with indeterminate QFT (2/14, 14.3%) compared to those with conclusive QFT results (22/373, 5.9%; p=0.033).

Of the 373 participants with conclusive QFT results, 33.2% (124/373) had evidence of TBI (Table 1). Females constituted 57.6% (215/373) of the cohort, and 64.9% (242/373) were between the ages of 20 and 49 years (Table 1) with an average age of 34.6 years (range 8–80). About one in five (20.6%) had no education, 33.5% were unemployed, and 35.4% were wage earners. HIV infection was reported by 11.1% (38/341) among those with a known HIV status, and 17.7% (66/373) of participants had a positive malaria test by microscopy or malaria rapid diagnostic test (a positive pLDH or HRP2 band). At enrolment, four (1.1%) were asymptomatic, and 22 (5.9%) reported severe COVID-19 at enrolment or at any time during the fourteen days after diagnosis. There were five participants who experienced moderate to severe symptoms and were hospitalized within 0–7 days of their COVID-19 diagnoses. Only 9 participants did not complete the daily FLUPro+ questionnaires, indicating a 2.4% loss to follow-up rate (a median of 11 days missing, range 1–13).

Generally, self-reported COVID-19 severity increased with age; self-reported severe COVID-19 was observed in 14% of enrollees ≥50 years of age (Fig 2A). TBI prevalence steadily increased by age group, as expected in a TB-endemic setting, reaching a maximum of 57% in 40–49 year-olds (Fig 2B). Similarly, HIV prevalence increased with age reaching a

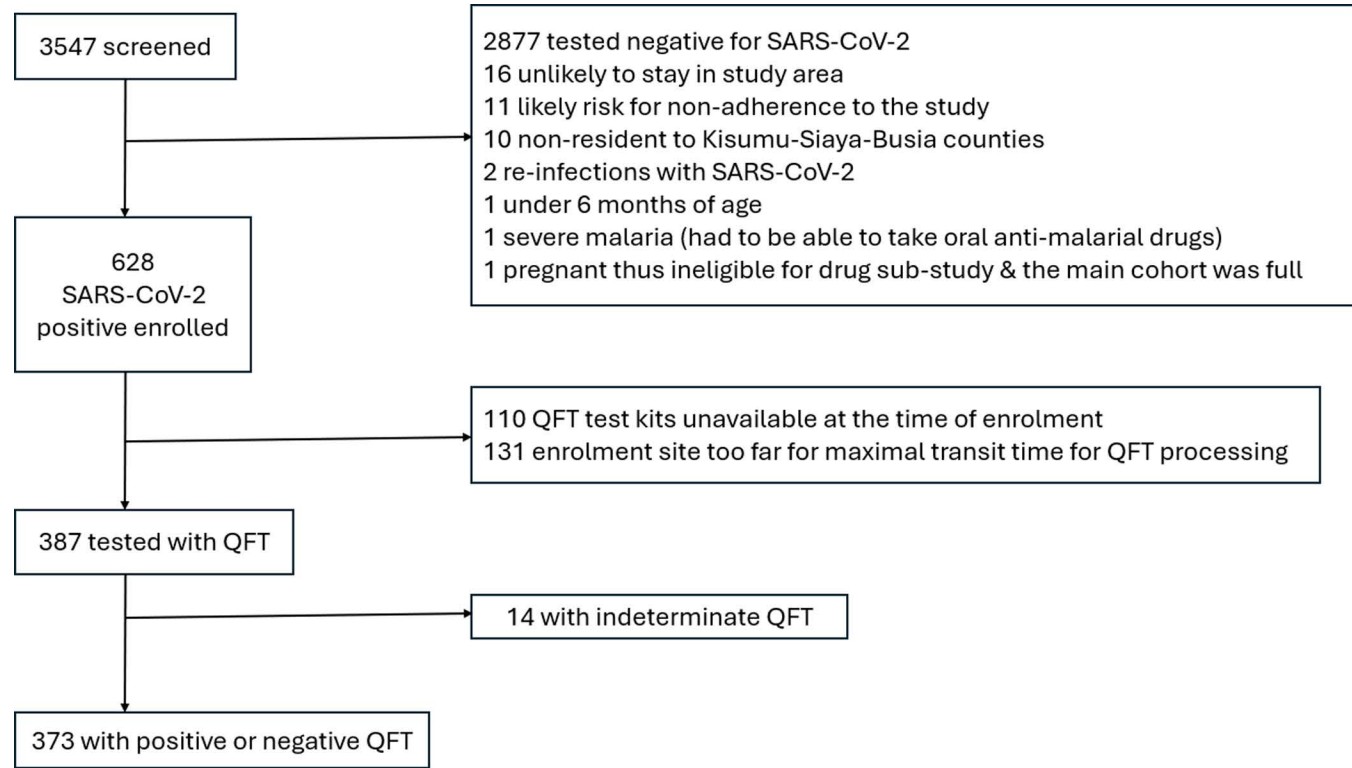

**Fig 1. Flow diagram of the MALCOV-QFT cohort, western Kenya, January 2021-January 2022. Note. QFT, QuantiFERON-TB Gold Plus test.**

maximum of 25% in 40–49 year-olds (Fig 2C). Among participants with malaria, 30% (20/66) were QFT positive compared to 34% (104/305) without malaria (odds ratio [OR] 0.84, 95%CI 0.47-1.49, p = 0.553).

Although self-reported severe COVID-19 was higher in persons with TBI than without TBI (Table 1), this was not statistically significant (8.1% [10/124] versus 4.8% [12/249]), OR 1.73, 95%CI 0.71-4.13, p = 0.215). There was little observable difference in the self-reported severity of COVID-19 during the 14 days of follow-up in persons with and without TBI (Fig 3).

Univariate analysis of participant characteristics indicated a statistically significant association of self-reported severe COVID-19 with age ≥ 50 years and HIV infection, while two other known risk factors – diabetes and hypertension – were of borderline significance (Table 2). Since the prevalence of TBI, a positive HIV infection status, diabetes, and hypertension all increase with age, several multivariable models were constructed to determine independent risk factors for self-reported severe COVID-19. The Akaike information criterion was used to identify the most parsimonious model; only age ≥ 50 years remained significant with an adjusted odds ratio of 2.91 (95% confidence intervals 1.02-7.69).

The median follow-up of the participants was for 105 days (range 0–368). After a follow-up of 152.5 person-years, there was one TB case and one death. Dexamethasone was administered intravenously or orally for 4–25 days (median 5 days) to seven participants. Although five of these seven had TBI, none developed active TB during the follow-up period that ranged from 198–368 days (median 328 days). The participant who died was a 57-year-old male with TBI, malaria parasitemia, diabetes, and hypertension; he died of severe pulmonary disease on the third day after being diagnosed with COVID-19. The participant with active TB was a 31-year-old male, who was QFT positive and had moderate COVID-19 from which he recovered without receipt of any anti-inflammatory drug. He was diagnosed with GeneXpert by a government facility at 128 days after being diagnosed with COVID-19 and was treated successfully with anti-TB therapy.



**Table 1. Baseline characteristics of SARS-CoV-2 polymerase-chain-reaction-positive study participants by TB infection (TBI) status, western Kenya, January 2021-January 2022.**

| Characteristic | | n (%) | TBI status* | | |
|---|---|---|---|---|---|
| | | | Positive n(%) | Negative n(%) | P‡ |
| Overall | | 373* (100.0) | 124 (33.2) | 249 (66.8) | – |
| Age (years) | <20 | 67 (18.0) | 7 (5.6) | 60 (24.1) | ref |
| | 20–49 | 242 (64.9) | 100 (80.6) | 142 (57.0) | **<0.001** |
| | ≥50 | 64 (17.2) | 17 (13.7) | 47 (18.9) | **0.017** |
| Sex | Female | 215 (57.6) | 68 (54.8) | 147 (59.0) | 0.439 |
| Education | None | 77 (20.6) | 14 (11.3) | 63 (25.3) | ref |
| | Completed primary school | 79 (21.1) | 20 (16.1) | 59 (23.7) | 0.281 |
| | Completed secondary school | 71 (19.0) | 30 (24.2) | 41 (16.5) | **0.001** |
| | Tertiary training | 146 (39.1) | 60 (48.4) | 86 (34.5) | **<0.001** |
| Occupation | Unemployed | 125 (33.5) | 20 (16.1) | 105 (42.2) | ref |
| | Wage earner | 132 (35.4) | 59 (47.6) | 73 (29.3) | **<0.001** |
| | Sales | 56 (15.0) | 28 (22.6) | 28 (11.2) | **<0.001** |
| | Subsistence farming | 29 (7.8) | 8 (6.5) | 21 (8.4) | 0.145 |
| | Other | 31 (8.3) | 9 (7.3) | 22 (8.8) | 0.095 |
| Hypertensive | | 32 (8.6) | 15 (12.1) | 17 (6.8) | 0.083 |
| Diabetic | | 9 (2.4) | 7 (5.6) | 2 (0.8) | **0.004** |
| History of TB | | 1 (0.3) | 0 (0) | 1 (0.4) | 0.473 |
| COVID-19 symptoms§ | No symptom | 4 (1.1) | 2 (1.6) | 2 (0.8) | ref |
| | Mild | 110 (29.5) | 30 (24.2) | 80 (32.1) | 0.320 |
| | Moderate | 237 (63.5) | 82 (66.1) | 155 (62.2) | 0.522 |
| | Severe | 22 (5.9) | 10 (8.1) | 12 (4.8) | 0.866 |
| HIV-infected§ | | 38 (10.2) | 14 (11.3) | 24 (9.6) | 0.821 |
| Malaria-infected | | 66 (17.7) | 20 (16.1) | 46 (18.5) | 0.553 |

*Excludes 14 individuals with indeterminate QuantiFERON Gold Plus test results.

§Self-reported COVID-19 severity used the validated FLUPRO+ tool. HIV status was also self-reported; 56% (18/32) of unknown HIV participants were children <18 years of age; excluding HIV unknowns, 11.7% of TBI+ were HIV + and 10.9% of TBI- were HIV + .

‡Statistically significant p-values are bolded.

## Discussion

In a prospective cohort of SARS-CoV-2 infected persons (98.9% symptomatic) conducted in a setting where TB, malaria and HIV are prevalent, we did not find an association between TBI and self-reported severe COVID-19. The main risk factor for self-reported severe COVID-19 was increasing age, as has been demonstrated in multiple studies worldwide [21,22]. Additionally, although it must be interpreted cautiously, we did not find evidence to support an increased risk of active TB among persons with TBI who almost exclusively had symptomatic COVID-19. Compared to the placebo arm of a 2014–2015 trial that enrolled QFT-positive, HIV-uninfected adult Africans (including in Kenyans), we observed a similar active TB incidence of 0.7 versus 0.6 per 100 person-years [23].

T cell immunity appears to be a key protective factor in the defense against severe COVID-19 [24]. In one study of patients with COVID-19 who had either TB disease, TBI or neither, the anti-SARS-CoV-2 spike protein T cell response in those with TBI and those without TBI was no different [25]. The similarity in the T cell immune response against SARS-CoV-2 in persons with and without TBI is consistent with our observation that TBI – as detected by the T cell-based QFT assay – does not predispose to self-reported severe disease.

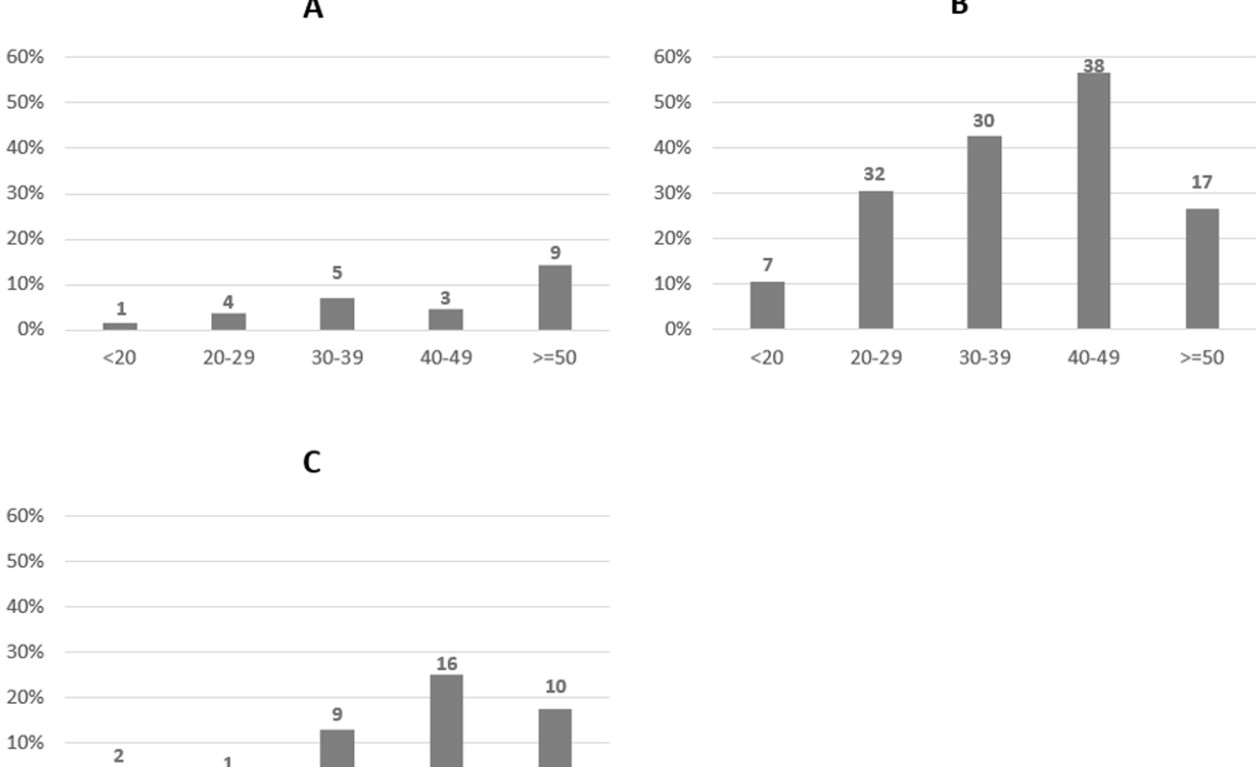

**Fig 2. Proportions of self-reported severe COVID-19, TB infection and HIV prevalences in SARS-CoV-2 infected persons by age group, western Kenya 2021-2022 (N =373).** Note. (A) Severe COVID-19 during 14 days after diagnosis (n = 373); (B) TB infection (n = 373); (C) HIV infection (n = 341); age groups are in years and numbers are shown above the bars.

From another perspective, it has been surmised that persons with TBI may be protected from severe COVID-19 based on an ecologic analysis showing an inverse association between severe COVID-19 and past epidemics of TB by country [13]. The authors suggested that "trained immunity" – the non-specific protection conferred by infection with one microorganism against illness from another microorganism – was responsible for this effect. In view of our non-significant trend to more severe disease with TBI (OR 1.73 95% CI 0.71,4.13), we did not find evidence supporting this hypothesis in our prospective cohort.

Several researchers have speculated that immune system dysregulation caused by COVID-19 could activate TB disease among persons with TBI [6,26,27]. This speculation has been supported by case reports [12]. The influenza virus is a well-established source of contagion with annual epidemics and regularly intersects with the global TB pandemic. A carefully conducted study of persons with active TB in Indonesia who had serologic evidence of prior influenza infection did not find that influenza infection predisposed to active TB [28]. While evidence for increased rates of COVID-19 death in persons with active TB or recent past TB exists [4], the hypothesis that COVID-19 could trigger active TB in persons with TBI remains unproven and is not supported in our prospective cohort.

Yet another group of researchers have suggested that persons with severe COVID-19 and TBI who have received corticosteroid therapy may be at risk of developing active TB [6,10,29]. We did not find evidence to support this hypothesis, possibly due to low numbers of study participants who required corticosteroid therapy. Nevertheless, for a long time

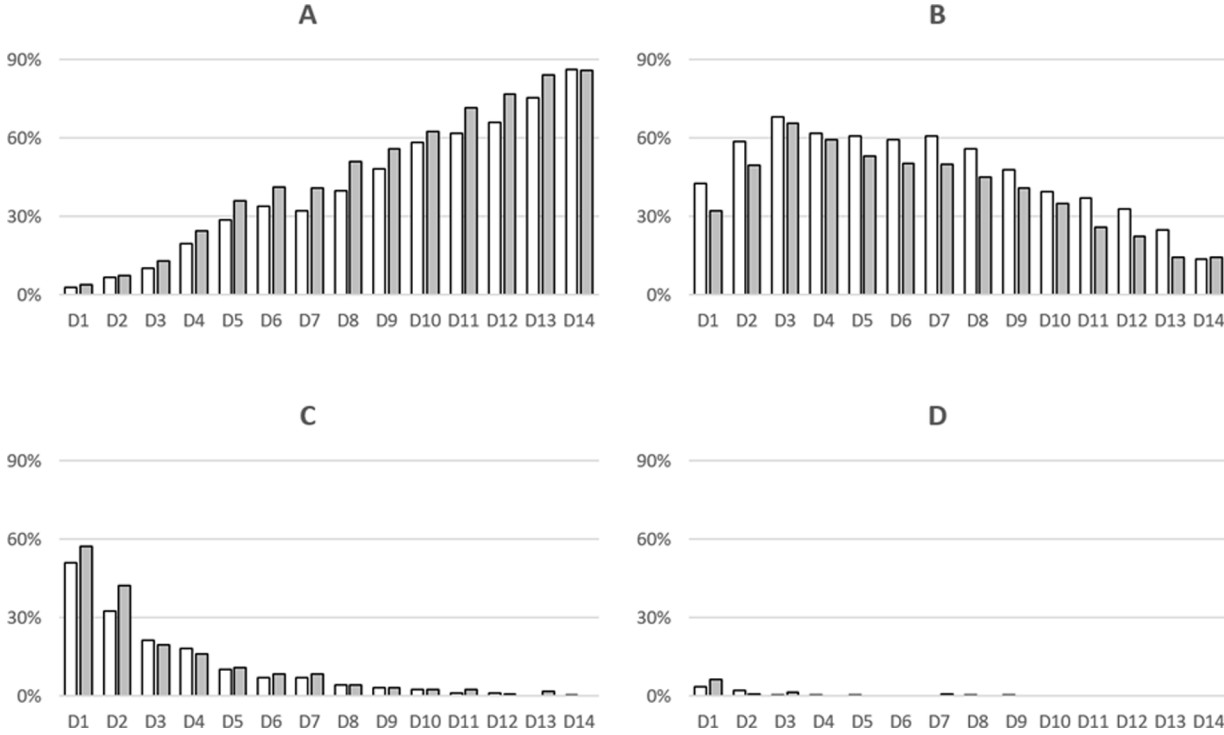

**Fig 3. Self-reported COVID-19 illness by severity and TB infection (TBI) status over two weeks, western Kenya 2021-2022 (n = 373).** Note. (A) No symptoms; (B) Mild illness; (C) Moderate illness; (D) Severe illness. Open bars = QuantiFERON Gold Plus-negative, grey bars = QuantiFERON-TB Gold Plus-positive (i.e., with TBI).

**Table 2. Risk factor analysis for self-reported severe COVID-19 in the MALCOV cohort, western Kenya 2021-2022 (n = 373).**

| Characteristic | | Severe COVID-19 | | Non-severe COVID-19 | | Crude Odds Ratio | | Adjusted Odds Ratio‡ | |
|---|---|---|---|---|---|---|---|---|---|
| | | n | % | n | % | OR (95%CI) | P | aOR (95%CI) | P |
| Age | ≥ 50 years | 9 | 14.0 | 55 | 85.9 | 3.73 (1.47, 9.07) | 0.004 | 2.91 (1.02,7.69) | 0.035 |
| Sex | Female | 12 | 5.6 | 203 | 94.4 | 0.87 (0.37, 2.12) | 0.762 | – | |
| Education | >primary | 12 | 5.6 | 202 | 94.4 | 0.89 (0.37, 2.1) | 0.782 | – | |
| Employed | | 16 | 6.5 | 232 | 93.5 | 1.37 (0.55, 3.89) | 0.524 | – | |
| Hypertensive | | 4 | 12.5 | 28 | 87.5 | 2.56 (0.70, 7.43) | 0.110 | – | |
| Diabetic | | 2 | 22.2 | 7 | 77.8 | 4.91 (0.70, 22.0) | 0.056 | – | |
| TBI* | | 10 | 8.0 | 114 | 91.9 | 1.73 (0.71, 4.13) | 0.215 | – | |
| HIV-infected† | | 5 | 13.2 | 33 | 86.8 | 3.13 (0.96, 8.77) | 0.039 | 2.77 (0.84, 7.93) | 0.070 |

*QuantiFERON-TB Gold Plus positive

†Self report

‡The final model included only those variables that had a p < 0.05 in univariate analysis. Multivariable logistic regression models that included variables with p < 0.1 (age ≥ 50, diabetes, HIV) or p < 0.2 (four variables) or p < 0.3 (five variables which included TBI) showed no variable to be statistically significantly associated with self-reported severe COVID-19.

clinicians have been aware of the heightened risk of active TB among persons receiving glucocorticoid therapy [30]. While the American Thoracic Society and CDC 1999 guideline recommends testing for TBI and potentially treating persons for TBI who receive "a prolonged course of high-dose corticosteroid therapy,"[31] the vast majority of severe COVID-19 patients receive a relatively short duration of steroid therapy as observed in our cohort. Additional research is needed to better determine the risk of active TB among persons with TBI who had received corticosteroids or other forms of immune suppressive therapy for severe COVID-19.

Interestingly, among participants with indeterminate QFT results, there was a statistically higher proportion with self-reported severe COVID-19 than among those with a conclusive QFT result. This has been noted by other researchers as well [32–34]. The explanation given is that due to the immune dysregulation and lymphopenia caused by severe COVID-19, interferon-release assays – assays that depend upon intact T cell responses – are less likely to yield definitive QFT results.

A legitimate concern affecting our setting is that co-infection with malaria may adversely affect the QFT assay. We did not observe an association between malaria infection and a positive QFT test. A team studying a cohort of Ugandan TB cases and their household contacts found that the QFT assay was unaffected by co-infection with either helminths or malaria [35].

Several limitations of our cohort study should be noted. Our convenience cohort may not have been representative of the general population. Firstly, asymptomatic SARS-CoV-2 is common in Kenya [36,37]. Therefore, our cohort's population of almost entirely symptomatic individuals likely represents a minority of persons co-infected with SARS-CoV-2 and TBI in western Kenya. Secondly, the study was underpowered to determine whether the 3% absolute difference in self-reported severe COVID-19 between persons with and without TBI was statistically significant. However, even if we had observed a significant difference, the high correlation of TBI with age would likely not have changed our conclusion in a multivariable analysis. A third limitation is that while HIV co-infection is an established risk factor for severe COVID-19 [38], HIV status was self-reported and unknown in 8.6% of our cohort, so we could not establish whether it was an independent risk factor for self-reported severe COVID-19. Our multivariable analysis that included two variables – HIV and age ≥ 50 years – showed that age was the main risk factor. A large study of hospitalized HIV-infected patients from around the world found that patients aged 45–75 years had increased odds of severe COVID-19 which again emphasizes the importance of age [39]. Hypertension and diabetes were self-reported conditions and so – just as with HIV status – we cannot exclude a risk of a misclassification error for these variables. It should also be noted that while clinical severity scales typically measure the impact of the disease on objective health outcomes, symptom severity scales such as FLUPRO+ capture patients' subjective experience of the illness. Capturing patient-centered outcomes in clinical trials has become increasingly recognized. Finally, as we only had one death, it was not possible to determine risk factors for COVID-19 death. Strengths of our study include the prospective observation of a large cohort of TBI persons with their first documented SARS-CoV-2 infection, a 98.4% completion of follow-up during 14 days after lab-confirmed diagnosis and a patient-centered orientation which has proven to be valuable in empowering clinical trials participants [40].

## Conclusion

TBI, as measured by the QFT assay, was not associated with self-reported severe SARS-CoV-2 infection in a prospective cohort of SARS-CoV-2 infected persons in western Kenya. Age greater than or equal to 50 years was the only risk factor associated with self-reported severe COVID-19.

## Supporting information

**S1 File. Study protocol, "Malaria as a risk factor for COVID-19 in western Kenya and Burkina Faso (MALCOV) ".**
(PDF)



**S2 File. Primary study manuscript pre-print: Barsosio et al. "Malaria as a risk factor for COVID-19 in western Kenya and Burkina Faso (MALCOV): an observational cohort study ".**
(DOCX)

## Acknowledgments

During a time of uncertainty and fear, we deeply appreciate the contributions made by the study participants who consented to join the MALCOV study, and the community health volunteers who were on the frontline recruiting these participants who were infected with a then much feared virus. We highly appreciate Dr. Patrick Munywoki's permission to use unpublished data in this manuscript regarding the seroprevalence of SARS-CoV-2 in public health surveillance sites in Kenya.

## Author contributions

**Conceptualization:** Taraz Samandari, Hellen C. Barsosio, Joshua Ongalo, Kimberly D. McCarthy, Simon Kariuki, Aaron M. Samuels, Chris Drakeley, Feiko O. ter Kuile.

**Data curation:** Taraz Samandari, Brian Tangara, Kephas Otieno, Maia Lesosky.

**Formal analysis:** Taraz Samandari, Luke Hannan, Maia Lesosky.

**Funding acquisition:** Taraz Samandari, Anand Date, Aaron M. Samuels, Chris Drakeley, Feiko O. ter Kuile.

**Investigation:** Brian Tangara, Joshua Ongalo, Morine Achieng, Kimberly D. McCarthy, Miriam Wanjiku, Julian Matthewman.

**Methodology:** Brian Tangara, Joshua Ongalo, Tegwen Marlais, Kimberly D. McCarthy, David Allen.

**Project administration:** Taraz Samandari, Hellen C. Barsosio, Brian Tangara, Tegwen Marlais, Kimberly D. McCarthy, Simon Kariuki, Aaron M. Samuels, Chris Drakeley, Feiko O. ter Kuile.

**Supervision:** Hellen C. Barsosio, Brian Tangara, Simon Kariuki, Chris Drakeley, Feiko O. ter Kuile.

**Visualization:** Taraz Samandari, Luke Hannan, Maia Lesosky.

**Writing – original draft:** Taraz Samandari, Hellen C. Barsosio, Feiko O. ter Kuile.

**Writing – review & editing:** Taraz Samandari, Hellen C. Barsosio, Brian Tangara, Joshua Ongalo, Kimberly D. McCarthy, David Allen, Anand Date, Maia Lesosky, Feiko O. ter Kuile.

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
