## [Decision Letter · Decision Letter 0]

2 Dec 2024

PGPH-D-24-02424

Severity of COVID-19 among persons with tuberculosis infection in western Kenya, 2021

Dear Dr. Samandari,

Thank you for submitting your manuscript to PLOS Global Public Health. After careful consideration, we feel that it has merit but does not fully meet PLOS Global Public Health’s publication criteria as it currently stands. Therefore, we invite you to submit a revised version of the manuscript that addresses the points raised during the review process.

EDITOR:  

In this study, severe COVID-19 was defined using the FLU-PRO scale, a patient-reported outcome measure that assesses the severity of influenza-like symptoms across six domains: nose, throat, eyes, chest, gastrointestinal, and body/systemic symptoms. This approach differs significantly from the standard World Health Organization (WHO) ordinal clinical severity scale and other clinical scales, which typically define severe COVID-19 based on clinical outcomes such as hospitalization, admission to the intensive care unit (ICU), intubation or mechanical ventilation, or death. By utilizing the FLU-PRO scale, the authors have focused on self-reported COVID-19 symptom severity rather than objective clinical indicators of disease progression.

To accurately reflect this distinction and prevent potential misinterpretation, the title, abstract, and main text need revision and replacing of "COVID-19 severity" with "COVID-19 symptom severity." The Discussion section also requires addressing this important differentiation, emphasizing that while clinical severity scales measure the impact of the disease on health outcomes, symptom severity scales capture the patient's subjective experience of the illness. Recognizing the difference between these concepts is crucial for interpreting the study findings.

We look forward to receiving your revised manuscript.

Kind regards,

María Elvira Balcells, M.D., MSc

Academic Editor

Journal Requirements:

Additional Editor Comments (if provided):

1. Introduction: Provide a brief description of the parent MALCOV study design for context.

2. Screening and Recruitment: The strategy used to screen and recruit participants for this cohort is unclear and requires elaboration.

3. Exclusion Criteria: Were there any exclusion criteria? If so, please specify. Additionally, clarify why active TB was not considered an exclusion criterion.

4. Follow-Up: What was the follow-up strategy employed for this study? Please provide details.

5. SARS-CoV-2 PCR Follow-Up: Clarify the relevance of regular SARS-CoV-2 PCR follow-up for this cohort in relation to the study objectives. If it is not relevant, consider removing it from the Methods section.

6. Parent Study: The NCT04695197 identifier corresponds to the parent study, not this one. Ensure this is clearly stated.

7. Sample Size Calculation: Include details of the sample size calculation to support the study's statistical validity.

8. Results: Please provide the mean age of participants along with the range, and also provide in outcomes the hospitalization proportion.

9. Style: Avoid starting sentences with numbers to maintain formal scientific writing standards.

Reviewers' comments:

Reviewer's Responses to Questions

**Comments to the Author**

1. Does this manuscript meet PLOS Global Public Health’s publication criteria? Is the manuscript technically sound, and do the data support the conclusions? The manuscript must describe methodologically and ethically rigorous research with conclusions that are appropriately drawn based on the data presented.

Reviewer #1: Partly

Reviewer #2: Partly

Reviewer #3: Yes

2. Has the statistical analysis been performed appropriately and rigorously?

Reviewer #1: No

Reviewer #2: Yes

Reviewer #3: Yes

3. Have the authors made all data underlying the findings in their manuscript fully available (please refer to the Data Availability Statement at the start of the manuscript PDF file)?

Reviewer #1: Yes

Reviewer #2: Yes

Reviewer #3: Yes

4. Is the manuscript presented in an intelligible fashion and written in standard English?

Reviewer #1: Yes

Reviewer #2: Yes

Reviewer #3: Yes

5. Review Comments to the Author

Reviewer #1: The study is interesting, given that current evidence supports the adverse effects of TB/SARS-CoV-2 co-infection or a history of active TB and COVID-19. However, there needs to be more evidence regarding the effect of TBI on the development of severe COVID-19; despite this, some issues should be considered to improve the clarity of the study and its scope.

The study presents a possible selection bias due to excluding people from areas with limited access to the health system (n=231), which limited access to the QTF test. These individuals could have experienced severe COVID-19 or died without adequate diagnosis, affecting the representativeness of the sample. It would be helpful to compare the baseline characteristics of the included and excluded participants to detect possible systematic differences that could influence the findings. In addition, only 22 cases of severe COVID-19 were captured, which limits the statistical power and the representativeness of the study, and these aspects should be discussed in the limitations section. The possible loss to follow-up needs to be mentioned, which could introduce additional biases in the results, especially regarding mortality and development of active TB.

It is unusual that the only variable significantly associated with severe COVID-19 is age >50 years, as previous studies have also pointed to male sex and socioeconomic disadvantage as significant risk factors. This finding could be due to a selection bias derived from the convenience sampling described in the methodology. In addition, the reliance on self-reporting to assess comorbidities and outcomes increases the risk of misclassification bias. In particular, self-reporting of HIV may be underestimated due to the associated stigma, and it is not specified whether participants were immunocompromised or on antiretroviral treatment with controlled immunity. This lack of detail limits the interpretation of the impact of HIV on the results, so it is recommended to clarify this variable to improve the precision of its assessment (The methodology states that information on medications received by participants was accessed).

Since the pediatric population is at lower risk for severe COVID-19 and TBI, it would be beneficial to stratify the analysis by age groups (adults and children) to avoid a possible distortion in the magnitude of the association between TBI and severe COVID-19, as well as in the subsequent development of active TB after viral infection.

At first, it seems that only the association between TBI and severe COVID-19 as an outcome will be evaluated; however, given the prospective nature of the study, an attempt was made to evaluate the reactivation of TBI post-COVID-19, but there is no statistical power to support this analysis, therefore the conclusion that there is no evidence of the association between COVID-19 infection (at all levels of severity) and the risk of TB activation seems premature due to the substantial imbalance between the study groups: 369 cases of COVID-19 versus only four asymptomatic controls. This imbalance limits the statistical power of the analysis and could affect the robustness of the results, suggesting that it should be interpreted with greater caution.

As an ethical consideration, did the study evaluate the possibility of providing antibiotic prophylaxis to participants with evident risk factors for developing active TB, or were these participants referred to the health system for a medical decision regarding this prophylaxis?

To support the hypotheses raised, it is recommended to include recent meta-analyses on TB/SARS-CoV-2 co-infection, as they would enrich the introduction and strengthen the study's scientific basis. In addition, reference 12 suggests that TBI predisposes to severe COVID-19; however, the cited article documents a case of reactivation of TBI to active TB in a patient with severe COVID-19, which should be clarified to avoid confusion in interpretation.

Finally, the statistical analysis section should be expanded to detail how the assumptions of the logistic model are verified and to explain better the methods used. It would also be helpful to include diabetes and hypertension in the description of baseline characteristics since these variables were considered in the adjusted analyses and could influence the results.

It is necessary to include all additional information from the study in the supplementary material to clarify any doubts the reader may have and avoid misinterpretation of the evidence presented.

Reviewer #2: This manuscript reports results of a prospective study evaluating COVID-19 severity among people with and without TB infection (TBI) in western Kenya. The study’s results suggest that those with TBI are not at elevated risk of severe COVID-19. Study strengths include QFT testing to determine TBI status, use of a validated tool to assess COVID-19 severity, and follow-up of up to 18 months to assess longer-term outcomes. Key limitations include the non-random sample and high proportion of possible participants not tested via QFT. The manuscript currently lacks some detail that would help with interpretation of the results.

Major comments:

• The manuscript notably lacks information about COVID-19 vaccination in the study population, which would have an important impact on COVID-19 severity. Was data on vaccination collected?

• Because of the methods for recruitment, the study population may have been at increased risk of severe COVID-19 compared to the general population; the authors note this in the discussion. It may be helpful for the recruitment flow diagram to include details on how many participants were recruited via each approach (attending a COVID-19 test center; admitted to COVID-19 isolation/quarantine facility; contact of a case; tested due to high-risk occupation).

• Additional details about the FLU-PRO tool would be useful (perhaps in supplementary information). What symptoms are assessed, and how is severity rated?

• The demographic and clinical data presented are very limited. Were any other data collected? E.g., COVID-19 vaccination, COVID-19 history, TB history, ART status (for those with HIV), COVID-19 treatment received, etc.

• Did the multivariable regression account for COVID-19 treatment?

• The methods state that “additional follow-up occurred for up to 18 months,” however it is not clear how follow-up was performed, how frequently, or what data were collected. What proportion were lost to follow-up? Did this vary by TBI or COVID-19 severity?

Minor comments:

• Were participants with a positive QFT referred for TB preventive therapy?

• Figure 3 is a bit difficult to interpret – what does “proportion of daily self-reported COIVD-19 illness” mean? Is this just the proportion reporting symptoms?

Reviewer #3: The study is a useful addition the the literature highlighting the t\point that presence of latent tb infection is not associated with Covid-19 disease severity. The study has a considerable sample size. In future viral pandemics like covid, the results of these study will help us understand the relationship between TB infection and viral coinfection.

6. PLOS authors have the option to publish the peer review history of their article (what does this mean?). If published, this will include your full peer review and any attached files.

**Do you want your identity to be public for this peer review?** For information about this choice, including consent withdrawal, please see our Privacy Policy.

Reviewer #1: No

Reviewer #2: No

Reviewer #3: No

---

## [Editor Report · Decision Letter 1]

4 Feb 2025

PGPH-D-24-02424R1

Self-reported COVID-19 severity among persons with tuberculosis infection in western Kenya, 2021

Dear Dr. Samandari,

Thank you for submitting your manuscript to PLOS Global Public Health. After careful consideration, we feel that it has merit but does not fully meet PLOS Global Public Health’s publication criteria as it currently stands. Therefore, we invite you to submit a revised version of the manuscript that addresses the points raised during the review process.

We look forward to receiving your revised manuscript.

Kind regards,

María Elvira Balcells, M.D., MSc

Academic Editor

Additional Editor Comments (if provided):

Major comments:

• As already underlined by the reviewers, SARS-CoV-2 vaccine status is a key information regarding the interpretation of COVID-19 severity. The authors comment in their response that among the 387 sub-study participants reported in this manuscript, only five participants were known to have received a SARS-CoV-2 vaccine by the time of enrolment, which was also at the time of SARS-CoV-2 diagnosis. However, this information has not been incorporated in the revised manuscript results neither in the tables.

• As previously recommended in the first round of revisions. COVID-19 severity needs to be replaced by “COVID-19 symptoms severity or “self-reported severe COVID-19”. In the Abstract results paragraph, there are still three sentences that refer to “severe COVID-19” that need modified accordingly, as well as in the Abstract’s conclusion. This is particularly relevant as the terms are confusing when including in the same paragraph that a patient died from “severe COVID-19” (which is probably the only occasion in which this “severity” is not only a patient-reported symptom).

Methods:

• In the Methods section, inclusion criteria indicate age over 6 months. As this study evaluated patient reported outcomes, were the authors expecting to use a pediatric /adapted FLUPRO tool for young children? Or is there an error in the lower age limit for inclusion? Or is that the age-inclusion criteria for the parent study only? Please clarify.

Results:

• As previously recommended, please avoid starting sentences with numbers to maintain formal scientific writing (e.g. number “387” in first paragraph of Results section, and 57.6% in the third paragraph of Results section).

• Revise whenever referring to “COVID-19 cases” regarding the whole cohort, as those participants with asymptomatic SARS-CoV-2 infection are not a COVID-19 case (if no disease) (e.g.in the third paragraph of Results section)

• Results reporting for binary variables in Table 1 and Table 2 is a little bit redundant. I strongly recommend removing all unnecessary rows with duplicated information when reporting binary variables. This is the consensus for scientific journals, in general it is not necessary to include a row for female and another for male, HIV yes/not, hypertensive and not-hypertensive, etc (as the information is complementary), just choose one of them (usually the “yes”). Additionally, in Table 2, the p values column’s titles are missing.

• The authors report two cases of incident active TB for the cohort. However, one of them had already been diagnosed and was under TB treatment 1 month before acquiring the SARS-CoV-2 infection. Therefore, it is not correct to include this second case as a “new” TB case developing during the prospective cohort follow-up and I recommend removing that case from the analysis.

Discussion:

• In the Discussion, first paragraph: 1) this is not a prospective cohort of “COVID-19 cases”, as there were asymptomatic SARS-CoV-2 infection cases included. I would recommend replacing by a prospective “cohort of SARS-CoV-2 infections”.

• I recommend revising the order of the sentences in the Discussion, as it is presently a little bit “jumpy” between one subject and another. As an example: The last sentence of the 1st paragraph of the discussion would better read if introduced after the 1st sentence, as both relate to COVID-19 symptoms severity.

• As steroid use has a strong and well- established evidence as a risk factor to TB infection reactivation (https://pubmed.ncbi.nlm.nih.gov/16463407/) , I am little bit uncomfortable with the sentence related to reference °31 that may give the wrong idea that the risk is low, please rephrase.

• In the limitations paragraph, revise the second sentence which refers to “symptomatic” SARS-CoV-2 infection being common in Kenya. Shouldn’t’ this be “asymptomatic” SARS-CoV-2 infections?

• As the authors presently included in their manuscript capturing patient-centered outcomes in clinical trials has become increasingly recognized as very important, please add a reference.

Figures:

• The proportions of participants having TBI (+) in each age-group presented in Figure 2B does not seem to agree with the proportion and numbers of participants with TBI for each age group presented in Table 1. Please revise in detail.

• Figure 2: I recommend including the absolute numbers for each column (instead of repeating the information for percentages both on Y axis and above each column)
---

## [Editor Report · Decision Letter 2]

20 Feb 2025

Self-reported COVID-19 severity among persons with tuberculosis infection in western Kenya, 2021

PGPH-D-24-02424R2

Dear Dr Samandari,

We are pleased to inform you that your manuscript 'Self-reported COVID-19 severity among persons with tuberculosis infection in western Kenya, 2021' has been provisionally accepted for publication in PLOS Global Public Health.

Best regards,

María Elvira Balcells, M.D., MSc

Academic Editor